# Delay in Decision-Making Affecting Construction Projects: A Sustainable Decision-Making Model for Mega Projects

**Shabir Hussain Khahro** [1,*], **Haseeb Haleem Shaikh** [2], **Noor Yasmin Zainun** [2], **Basel Sultan** [1,*] **and Qasim Hussain Khahro** [2]

1   Department of Engineering Management, College of Engineering, Prince Sultan University, Riyadh 11586, Saudi Arabia
2   Jamilus Research Center, Faculty of Civil Engineering & Built Environment, Universiti Tun Hussein Onn Malaysia, Parit Raja, Batu Pahat 86400, Malaysia
*   Correspondence: shkhahro@psu.edu.sa (S.H.K.); basel.sultan@psu.edu.sa (B.S.)

**Abstract:** The construction industry is one of the world's fastest-growing industries, since it directly and indirectly contributes to several other industries. It has been observed that delays in the decision making of important stakeholders negatively impact construction projects. Thus, this has multiple consequences on project performance. Consequently, the purpose of this study was to identify the primary causes of delayed decision-making and offer a decision support model for timely construction project decisions. For factor identification, a comprehensive literature analysis was conducted, followed by an analysis of questionnaire responses of ninety-one professionals. For data analysis, the relative importance index (RII) method was applied. The results indicate that client decision-making delays pose a substantial obstacle for building projects. The early decision-making process is impacted by a lack of technical competence, incomplete paperwork, poor leadership, and coordination/communication issues. The proposed model could also assist project practitioners in improving their project decision making. This research study encourages stakeholders to create and implement an efficient decision-making procedure for timely project decisions. A procedure for decision making that is successful could decrease delays in the decision-making process and prevent conflicts and disputes in projects.

**Keywords:** construction industry; decision making; delay; construction process; large-scale projects





## 1. Introduction

One of the major problems that the construction sector is reportedly experiencing today is decision-making delay. Project cost, timeline, quality of work, and a number of other elements are all impacted by late decision making. As the client is the only owner of the project, it is the client's major obligation to simplify the decision-making process in the project in order to make timely decisions. Project stakeholders are not adversely affected by delayed decision making at a high level [1]. If the client's decision-making process is delayed, it has an impact on project and delays the contractor's job. The customer may be faced with a greater number of claims to reimburse the contractor for extra time and costs in such a scenario [2].

Lack of organizational support, changes in market pricing, insufficient information from suppliers, and decision-making process unavailability are the main causes of decision-making delays [3]. Project delays and improper actions of clients, contractors, and consultants, among other important project players, have an impact [4]. It is reported that delay in the decision-making process is due to a lack of proper communication among the parties, lack of interest, lack of expertise, and lack of knowledge and information, resulting in serious project problems [5].

It is reported that the main reason for the delay in a construction project is the client's late decision-making process [6]. The client must improve decision-making efficiency for

the smooth completion of projects. Missing information and documents could create issues for stakeholders. The concerned stakeholders' delay in decision making may affect project performance. It could also create conflict between the owner and the contractor [7].

Previous research has suggested that the choice of decision-making strategies is highly related to the characteristics and actions of stakeholders in practice; however, only a limited number of studies have been conducted in the field of construction to investigate the significance of stakeholders' attributes, behaviors, and decision-making strategies. Construction is a field in which few studies have been conducted to analyze the importance of stakeholders' characteristics, attitudes, and decision making [8]. Construction projects are one-of-a-kind endeavors in terms of the design of a facility, as well as the organization of the project, the production facility, and the production procedures. Construction projects may also be unique in terms of the production processes themselves. They are distinguished by the presence of a large number of diverse stakeholders, which has historically acted as a barrier to the development of integrated information systems that call for the management of dispersed information and responsibilities [9]. Construction companies often fail to adopt a preventative approach when dealing with the inherent unpredictability of urban development projects. When difficulties emerge as a result of disregarding possible risks, it is typical for there to be delays in the project as well as an increase in the cost of the project. Inadequate information and inadequate management of project risks not only result in increased project costs, delays in project completion, and even the premature cancellation of the project before it is finished, but they also have a bad influence on the reputation of the project team [10].

Poor coordination and management at the project site may cause a delay in the decision-making process [11]. Additionally, it has been stated that poor communication among project participants poses a serious risk to large-scale infrastructure projects. To keep everyone updated on progress and any related difficulty, a robust and efficient coordination effort amongst the major stakeholders is essential [12]. One of the biggest issues that the construction industry is now experiencing is a lack of capable leadership that can handle, manage, and make prompt choices during large-scale infrastructure projects [13]. Construction companies need good and experienced leadership to make timely decisions and follow the decision process for smooth project execution [14].

Based on previous research, a comprehensive literature review was conducted with the goal of identifying the relevant factors. Table 1 provides an overview of the elements that have been emphasized in previous research as being factors that contribute to delays in decision making.

**Table 1.** Factors affecting delay in decision making in projects.

| Factors Affecting Delay in Decision Making | [15] | [16] | [17] | [18] | [19] | [20] | [21] | [22] | [23] | [11] | [24] | [25] | [26] | [27] | [28] | [29] | [30] | [31] | [32] | [33] | [34] | [35] | [36] | [37] | [38] | [39] | [40] |
|---|---|---|---|---|---|---|---|---|---|---|---|---|---|---|---|---|---|---|---|---|---|---|---|---|---|---|---|
| Inadequate staff to make decisions | √ | √ | √ | | | √ | √ | √ | √ | √ | | | | | √ | | | | √ | | | | | | √ | √ | |
| Technical expertise | | | | | √ | | | | | | | | | | √ | √ | | | | | | | | √ | | | |
| Political factor | √ | √ | | √ | √ | | | | | | | | | | √ | | | | | | | | | | | | |
| Lack of experience | | √ | √ | | | | | √ | | | | | | | | | | √ | √ | | √ | | | | | √ | √ |
| Poor project planning | | √ | | √ | | | | | | | √ | √ | | | | | | √ | | | | | | | | √ | √ |
| Lack of good leadership | | √ | | | | | | | | | | | √ | | | √ | | | | | | | | | | | √ |
| Shortage of time | | | | √ | | √ | | | | | | | √ | √ | √ | √ | √ | | | | | | | | | | |
| Mistakes in contract documents | | | | √ | | √ | | | | | | | | | | √ | | | | | | | | | | | |
| Lack of coordination/communication | √ | √ | | √ | | | | | √ | | | | √ | √ | | √ | | | √ | | | | √ | √ | | | √ |
| Financial problems | | | | | √ | | | | | | √ | | √ | | | | | √ | | | √ | | | | | √ | |

**Table 1.** *Cont.*

| Factors Affecting Delay in Decision Making | [15] | [16] | [17] | [18] | [19] | [20] | [21] | [22] | [23] | [11] | [24] | [25] | [26] | [27] | [28] | [29] | [30] | [31] | [32] | [33] | [34] | [35] | [36] | [37] | [38] | [39] | [40] |
|---|---|---|---|---|---|---|---|---|---|---|---|---|---|---|---|---|---|---|---|---|---|---|---|---|---|---|---|
| Irresponsibility of consultant | ✓ | | | | | | | | | | ✓ | | ✓ | | | | | | | | | | | | | | |
| Incomplete documents | | | ✓ | | ✓ | | ✓ | ✓ | | | | ✓ | | | | ✓ | | | | | | | | | | | ✓ |
| Poor management | | | ✓ | | | ✓ | | ✓ | ✓ | | | ✓ | | | | | | | | | | | | | | | ✓ |
| Irresponsibility of client | | | | | | | | | | | | | ✓ | ✓ | | | | | | ✓ | | | | ✓ | ✓ | | |
| Environmental and social factors | | | | | | | | | | | | | ✓ | ✓ | | | ✓ | | | | | | | | | | |
| Negotiation skills | | | | | | | | | | | | | | | | | | | | | | | | | | | ✓ |

Identifying errors in the early stage of a project can help resolve many issues of a project and assist the owner in making timely decisions [41,42]. Selecting an appropriate process in the early stages of construction is beneficial for all stakeholders [43,44]. It is recommended to adopt a decision support system to improve project performance and reduce conflict, dispute, and risk in the project [45].

The owner and the consultant should convey the decision to the contractor by the specified time mentioned in the contract and the decision support process. It increases project performance. A pre-construction assessment of project activities can assist in the timely completion of the project [33,46]. The contractor should perform some pre-project assessment of the project and identify the possible factors that could cause delays in the project. The contractor should inform the client regarding issues for discussion and finalize the issues before project execution [47,48]. The key project stakeholders are responsible for improving organization performance by making the right decision at the right time [49].

Given the significance of this particular topic, the purpose of this study was to determine the elements that are the most liable for delay in decision making in major fast-track building projects in Saudi Arabia. There have only been a few efforts in the past in this particular location, but the Saudi Arabian government is planning to start a large number of mega projects in the near future that will match the requirements of Saudi Vision 2030. In addition to this, this study offers a decision support model as a means of enhancing the decision-making process that is involved in building projects. This research study recommends several relevant critical metrics to aid decision makers in the construction sector in making timely judgments in projects. These measures are proposed in this study. This research could help stakeholders avoid delays in decision making and perform the necessary activities for timely and effective decision making in projects to ensure that they are completed on time, within budget, and to the required quality.

## 2. Decision-Making Process

Generally, any decision-making process is based on various components, as shown in Figure 1. It depends on the collection of data/information, which proposes various alternatives to choose the best alternative. A decision-making process is an organized way of making decisions with the support of relevant information gathered by various stakeholders on the project, including clients, consultants, and contractors. Decision makers should create a better environment to negotiate issues with possible alternative resolutions [50]. In the construction industry, there is a lack of trained practitioners and decision-making-relevant data. Owing to the deficiency of consolidated data, project decision makers struggle to make prompt judgments. The decision-making process generally used in the industry is shown in Figure 1 [51,52].

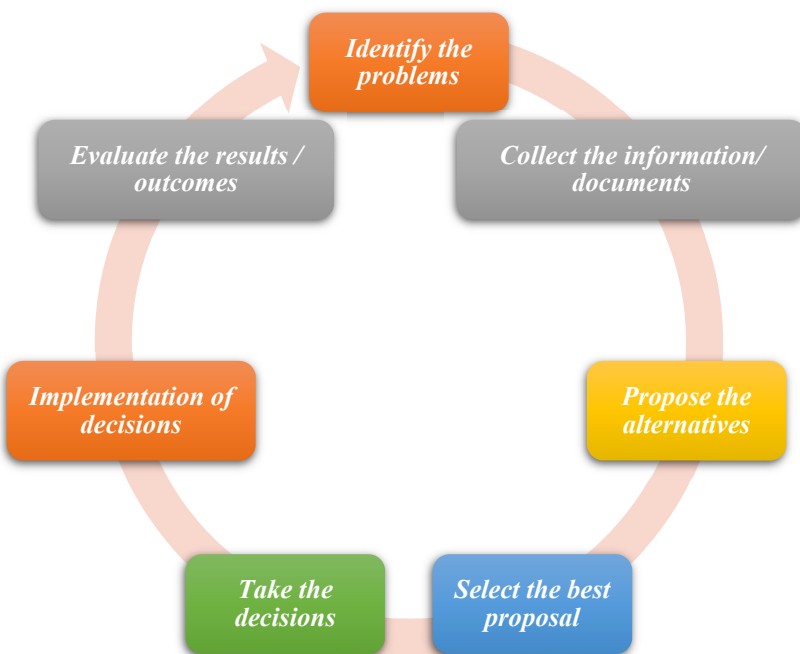

**Figure 1.** General decision-making process.

Prior to making any kind of conclusion, it is crucial to locate specific issues that have arisen during the project. For the purpose of finding a solution to the problem, the relevant data should be gathered. This includes communication data such as letters, evaluation reports of stakeholders, presentations, images, videos, and so on. These data assist those in charge of making decisions in making decisions that are suitable and correct. The person who makes decisions is responsible for coming up with the most beneficial ideas and other options for the project. After the selection of the most appropriate solution, the decision should be put into action in a timely manner in order to prevent conflict among the many stakeholders. The team that was chosen should perform regular evaluations of the decision that was taken in order to monitor its consequences and results in order to make decisions that are better informed in the future.

## 3. Literature Review on Previous Decision-Making Models

Various decision-making process could be recommended for facilitating and improving the decision process in any industry [53–55]. Various researchers conducted research on this important topic, and a few key concepts and models are presented in the present section of this paper [9,56,57].

The decision-making process is specifically separated into phases. Herbert Simon, an American physicist, published a decision-making model with four key stages in 1960:

(a) Initial stages of information gathering: problem description, primary goals, source information gathering, and comparison of the current condition and anticipated improvements.
(b) Decision modeling: analysis of gathered data, problem modeling, choice of standards, options, and techniques of decision making.
(c) Making decisions: executing tests and research, analyzing findings, and selecting the best option.
(d) Choice implementation: communicating the decision to implementers, determining if the best option was selected, carrying out the decision, and evaluating the outcomes, as illustrated in Figure 2.

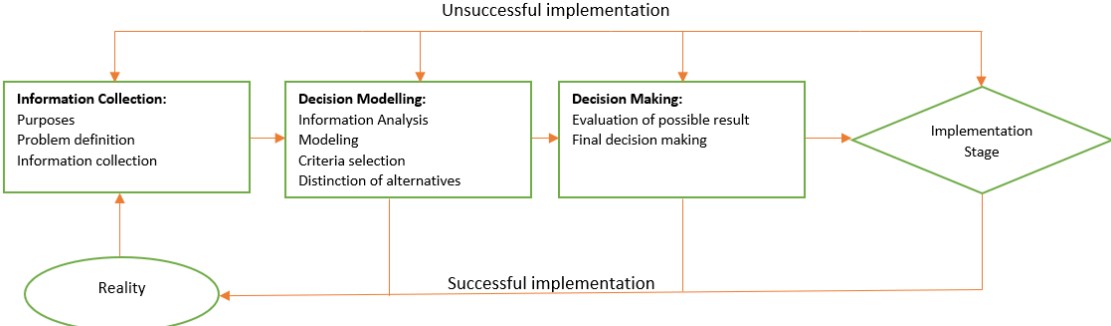

**Figure 2.** Decision-making stages [58].

The stated decision-making technique can be used in many scenarios, including energy-efficient renovations. This proposal is for energy-based initiatives, not building or construction projects. Projects may acquire and manage explicit and tacit knowledge using an efficient knowledge approach. "Explicit knowledge" is easily accessible within a company in the form of books and processes and may be kept for future use. People's ideas and organizational processes form tacit knowledge, which is high-level knowledge. Knowledge management requires balancing technology and soft elements, such as leadership, vision, strategy, incentive systems, and culture, to make information visible. The study results show that fundamental tacit knowledge features make it valuable in the building industry. Figure 3 shows the whole decision-making process and the importance of tacit knowledge in organizational performance and competitive advantage.

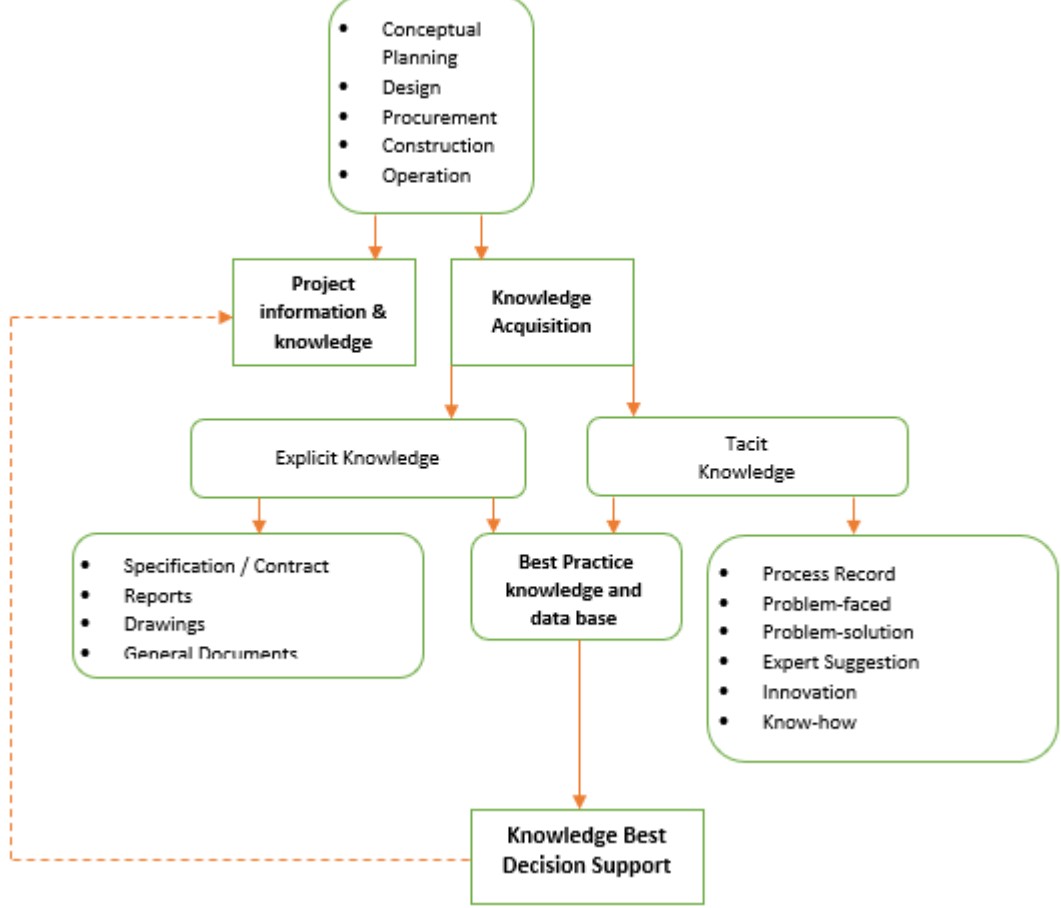

**Figure 3.** Knowledge-based decision support model [59].

Important issues throughout the construction stages are recognized by acquiring and using tacit and explicit knowledge. According to the authors, challenges of tacit knowledge include loss of experience, knowledge, problem-solving skills, and ingenuity. Explicit information challenges are usually caused by problems with data storage, since data may be partly or erroneously collected. Figure 4 depicts a suggested integrated conceptual framework for decision making to assist the implementation of IBS technology [60].

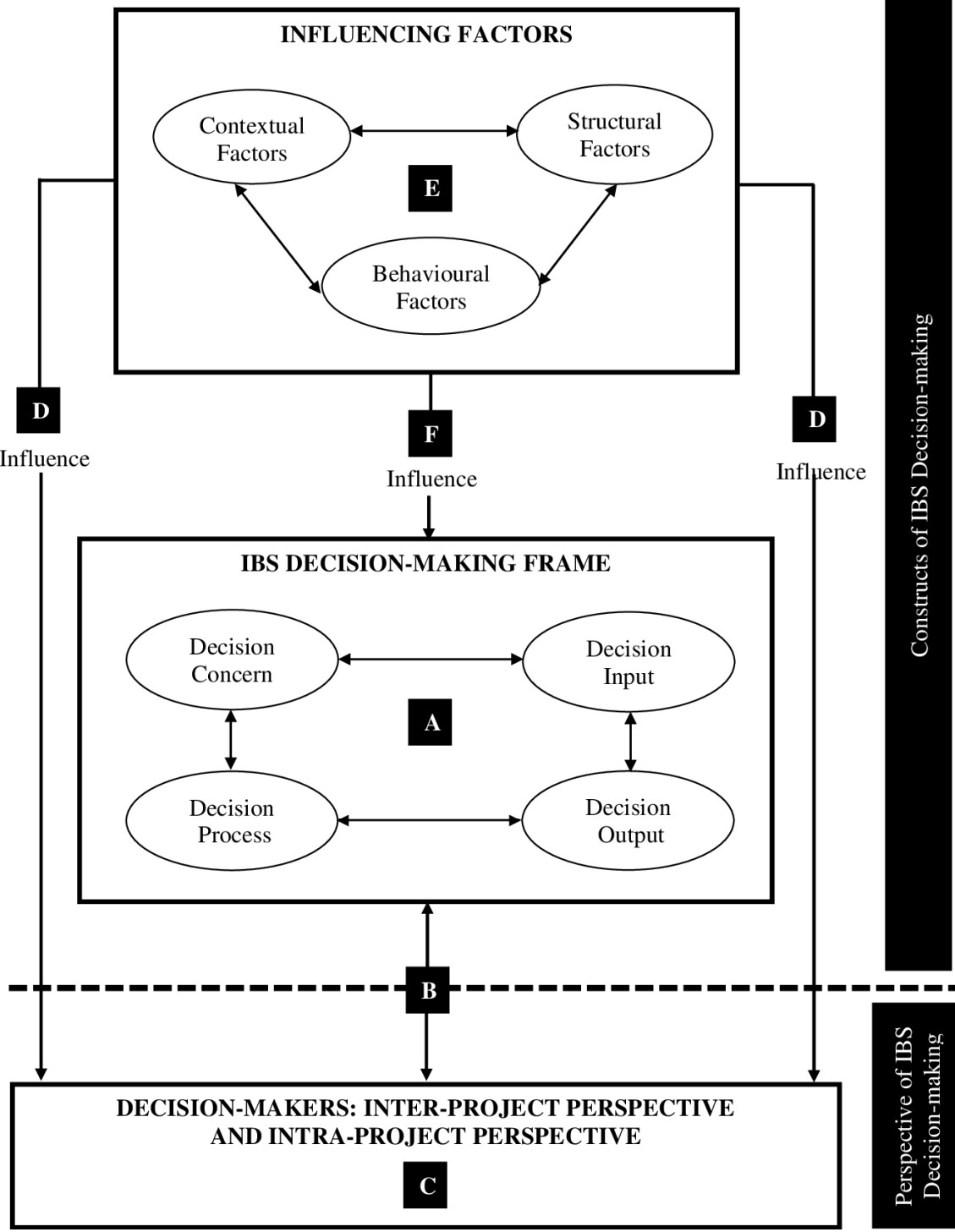

**Figure 4.** Project-based decision-making model [60].

Because of their interaction, environmental, structural, and behavioral aspects may have an influence not only on the IBS decision-making process, but also on decision makers from both inter- and intra-project perspectives, as shown in the figure. IBS decision making is dynamic rather than stable and linear in a decision-making framework because of interactions among numerous factors, such as "concerns," "inputs," "processes," and "outputs." These components can interact in a number of ways, resulting in different routes leading to different outcomes. The decision-making process utilized in construction projects, including the application of IBS technology, is typically influenced by the roles played by project participants and the decision paradigm selected for the project. These project participants examine IBS decision making and its components from a number of perspectives. An extra framework for making judgments on office building remodeling, i.e., a decision-making model for sustainable building refurbishment, emphasizing energy efficiency was created by [58] and is shown in Figure 5.

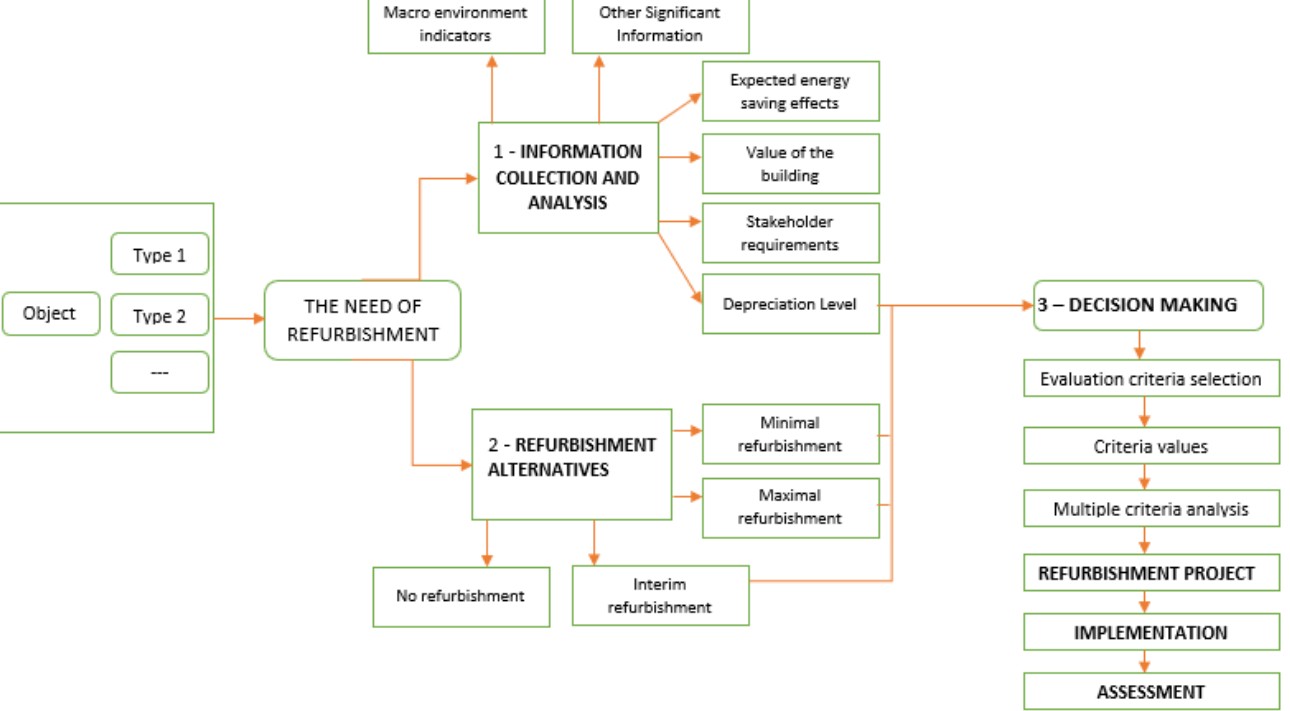

**Figure 5.** Decision-making model for sustainable buildings [58].

Six key steps make up the proposed decision-making paradigm. Each specific type of building and its surroundings are analyzed in the first step. Numerous indications and situations must be assessed, since in this model, energy-efficient renovation is considered from the perspective of sustainability.

The advantage of renovating a structure is typically viewed from the energy-saving perspective. Therefore, the predicted energy-saving impacts of renovation should be assessed. The potential for energy savings also affects the choices of renovation materials and alternatives. This technique presupposes that the relevance and priority of the versions under investigation are directly and proportionally dependent on a set of criteria that adequately describe the alternatives as well as the values and significances of the criteria. The authors' methodology for integrating stakeholders' economic, technical, social, and ecological concerns into decision making in sustainable building renovations aids in selecting the most energy-efficient renovation options by applying a variety of criteria and methodologies.

Based on a potential perspective on the components of a decision problem in 2015, along with their connections, Figure 6 illustrates the three fundamental components of

decision makers, decision tools, and selection strategies together with their interactions during the proposed decision-making process [61].

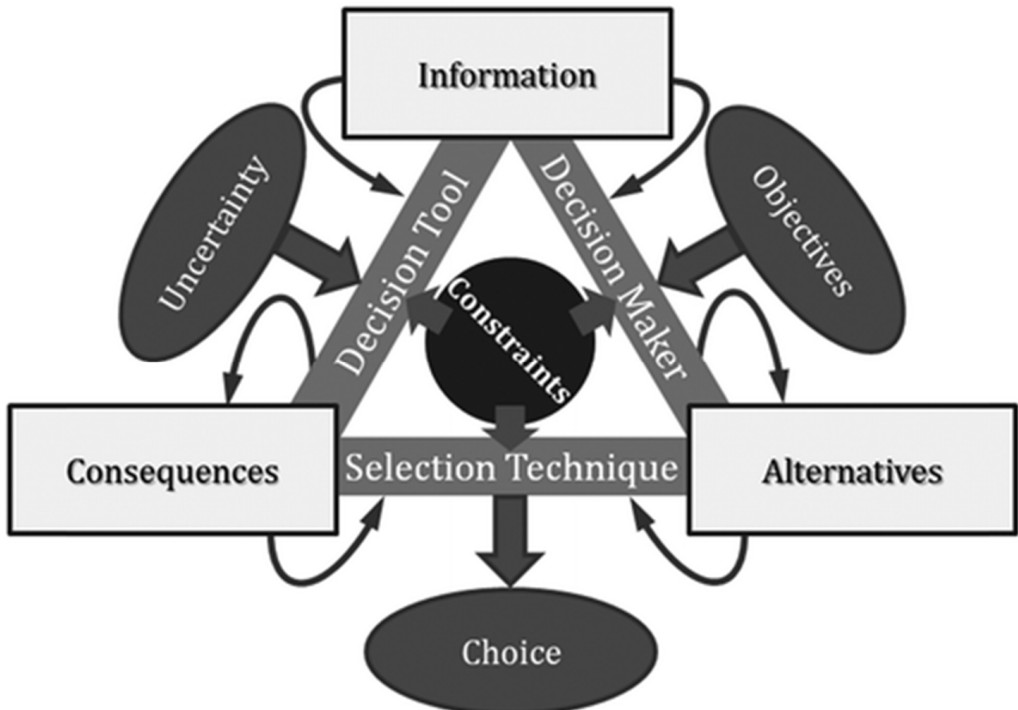

**Figure 6.** Some major components of decision-making problems [61].

This model attempts to highlight some of the important characteristics and components that have been successful in the development of the decision-making literature rather than attempting to offer the ontology of or a full model for decision making. Here, the literature on decision-making research within CEM is examined using this viewpoint. The terms "choice tool" and "selection technique," which may have been employed in various ways in other settings, should be distinguished from one another.

The architecture, administration, and combining of distributed application models in multi-models also represent a significant problem. One or more application models are employed across the engineering and managerial domains of the construction industry. The multidimensional information space of a project is defined by the total of its application models, with each domain standing for a different dimension. Data interchange between AEC/FM disciplines and project stages must be compatible in order for the project information from this model space to be reused. Additionally, data must be appropriately converted for use in decision making at various management levels of the project organization. As shown in Figure 7, the management levels of the owner and contractor organizations make up the primary decision hierarchy of a construction project.

As demonstrated by the two pyramids, the owner is more interested in the high-to-medium organizational levels, whereas the contractor is mostly interested in the medium-to-low organizational levels. As a result, cross-company information exchange may only occur at a few overlapping organizational levels, where interoperability must be concentrated. As a result, hierarchical information modeling is required for owner–contractor information integration.

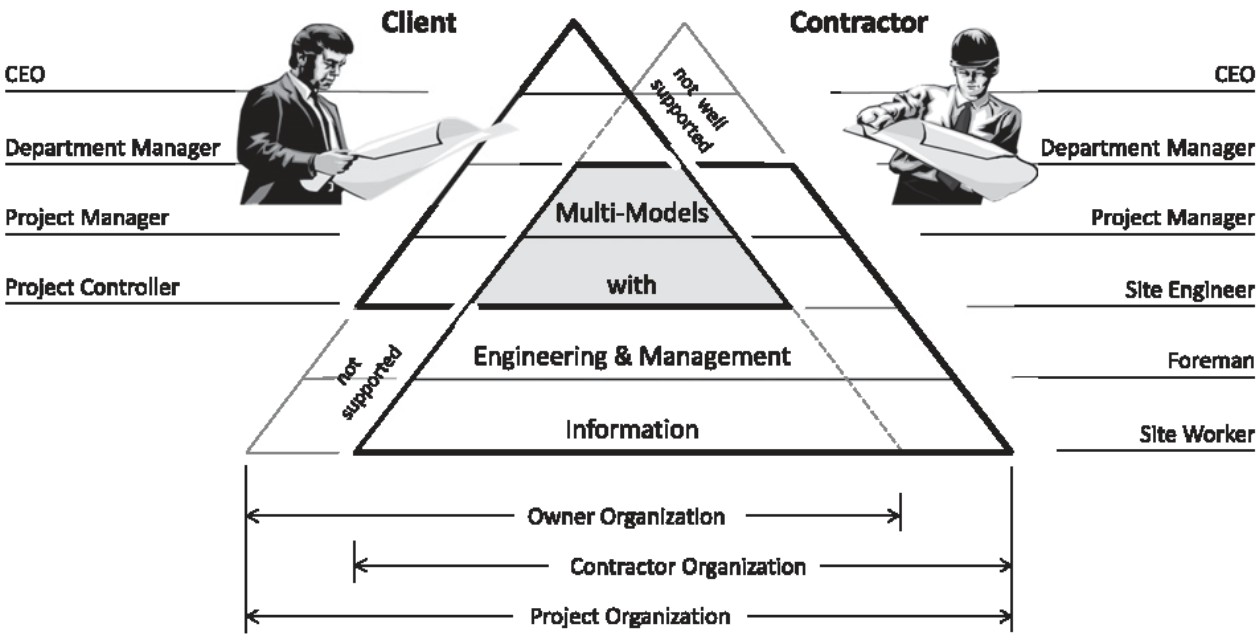

**Figure 7.** Decision hierarchies in a construction project [9].

This approach aims to define a model for creating different variations in building enclosures, a model for establishing the weight of criteria, a model for determining the initial weight of criteria (using expert methods), a model for the multi-variant design of building construction alternatives, a model for multiple criterion analysis and setting priorities, and a model for deciding the utility derivation of a project, as shown in Figure 8.

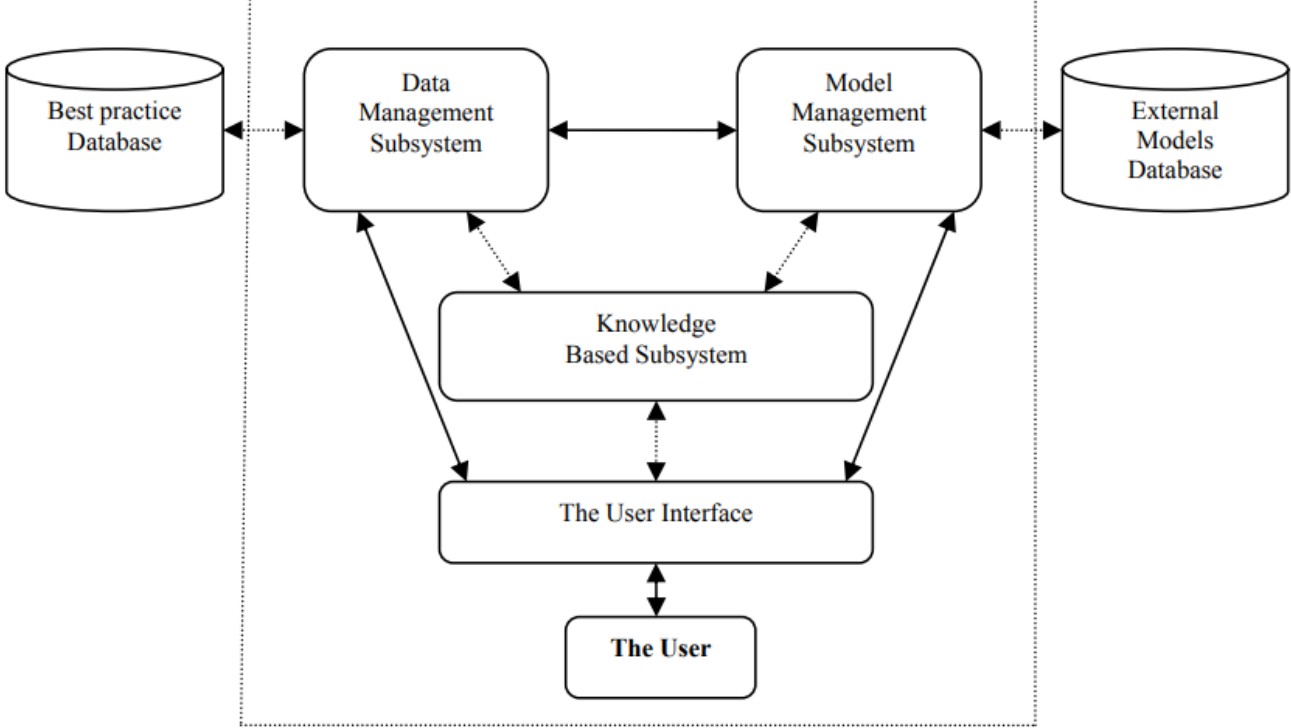

**Figure 8.** Architecture of knowledge-based decision support system [59].

To find the best construction options, the offered DSS-CP employs the complex proportional assessment technique. This approach assumes that the relevance and priority of the

versions under consideration are directly and proportionately dependent on a set of criteria that sufficiently specify the alternatives as well as the criterion values and significances. The values and initial significances of the criteria, as well as the system of criteria, are computed by experts. It was observed in the extensive literature review that there have been multiple research works on decision making. Most of the research works have focused on some specific conditions, and none of the existing models fits the construction industry scenario of Saudi Arabian infrastructure projects. Thus, this research aimed to create a decision-making model for infrastructure projects in general and specifically for construction projects in Saudi Arabia. This model could assist the stakeholders of the construction industry in Saudi Arabia in making informed and early decisions to finish projects on time, within budget, and meeting the quality standards.

## 4. Research Methodology

A detailed literature review was carried out for factor identification in existing research on delays and decision making. A questionnaire (Appendix A) was designed, and the feedback of experts was requested to analyze the problem. The data obtained from experts were collected using the questionnaire, and they were analyzed using the relative importance index. The results were validated using Cronbach's alpha as the reliability test.

The identified factors were transferred into the questionnaire, and it was sent to 125 experts in the construction industry for their feedback. The selected experts were project managers, engineers, and practitioners currently working on mega projects in Saudi Arabia and experts who had previous experience in managing large-scale projects. The questionnaire was distributed manually and sent via email to the experts. There were two parts to the questionnaire, as shown in Annexure I. Phase 1 of the questionnaire contained three sections. Section 1 contained the demographic information of the experts. In Section 2 of questionnaire, experts were requested to share their experience relative to stakeholder significance in the decision-making process. In Section 3 of questionnaire, factors that affect delay in the decision-making process were presented. The experts were requested to rank the key factors that cause delays in decision making in mega infrastructure projects. In Phase 2 of the questionnaire, the experts were requested to suggest possible solutions and share their feedback on the process for the timely decision making of stakeholders of projects.

## 5. Data Collection and Analysis

The final questionnaire was sent to 125 experts working on various mega construction projects in Saudi Arabia, including rail projects, BRT projects, and airport projects, and 91 questionnaires were successfully received. The data were analyzed using the relative importance index (RII) method. The RII method was successfully used in past research to achieve the ranks of such data sets [62]. Table 2 shows the comparison of various ranking methods used in previous studies.

**Table 2.** Ranking methods used in previous research.

| Ranking Method | [15] | [16] | [18] | [19] | [63] | [64] | [11] | [65] | [23] | [66] | [67] | [27] | [28] | [68] | [69] | [34] | [35] | [70] | [71] |
|---|---|---|---|---|---|---|---|---|---|---|---|---|---|---|---|---|---|---|---|
| Average index | | | √ | | | | | | | | √ | | | | | | √ | | |
| Mean value | √ | | | √ | | | | | | √ | | √ | √ | √ | | √ | | | |
| RIW | | | | | | | | | | | | | | | | | | | |
| RII | | √ | | | | √ | √ | √ | √ | | | | | | | | | √ | √ |
| Multiple regression | | | | | √ | | | | | | | | | | √ | | | | |

The RII assigns a weight to the respondent perception feedback. The RII formula used in this research is shown in Equation (1).

$$\text{RII} = \sum \frac{W}{A * N} \tag{1}$$

where $W$ = weightage given to each factor by the respondents, $A$ = weight (i.e., 1 to 5 in this case) and $N$ = the total number of respondents.

*Reliability Test*

The reliability test was conducted to analyze the consistency of the data gathered from the respondents. This research used Cronbach's alpha test using SPSS. Cronbach's alpha value of seven and above is a good fit; eight and above is a better fit; and nine and above is the best fit [72].

## 6. Results and Discussion

The experience of participants is a critical component in the decision-making process for such statistical issues. As a consequence of this, this aspect was taken into consideration when the data were collected. The surveys were sent to top specialists working on a variety of mega projects in Saudi Arabia. The total number of years the respondents had spent working in the construction industry is shown in Figure 9.

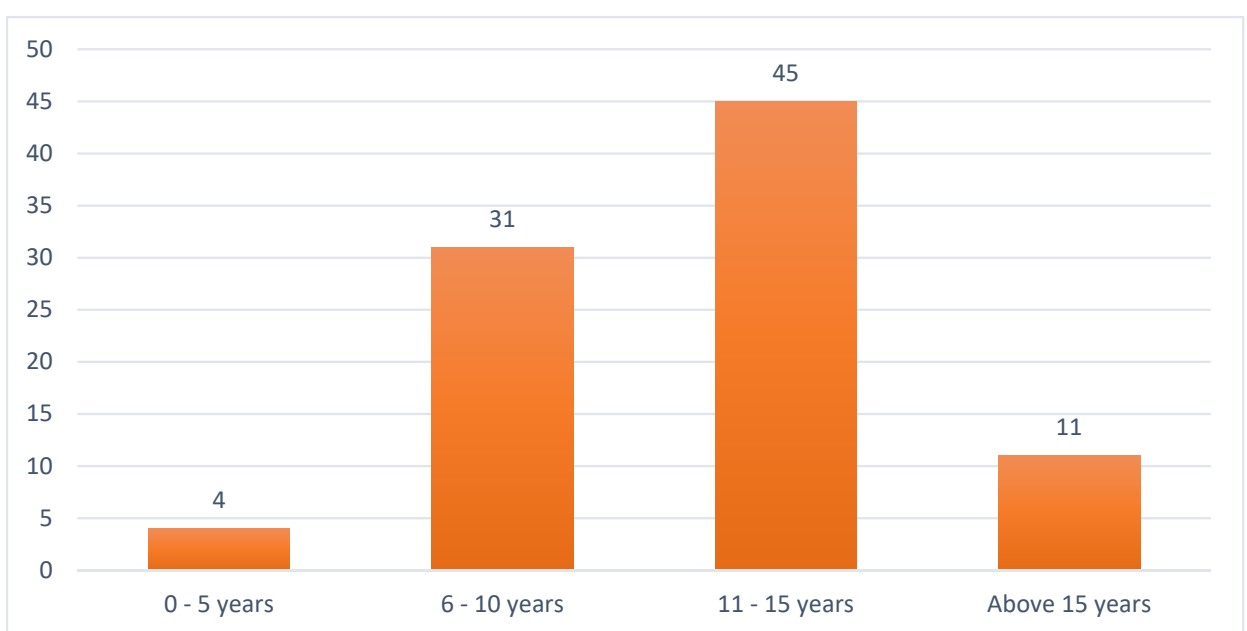

**Figure 9.** Respondents' years of experience.

Only 4 respondents had experience between 0 and 5 years, whereas 31 respondents had experience between 6 and 10 years; a total of 45 respondents, 11–15 years; and 11 respondents, more than 15 years.

As discussed above, following demographic information, the respondents were requested to share their experience relative to the key stakeholder responsible for delay in decision making in construction projects. Figure 10 shows the stakeholders responsible for delayed decision making.

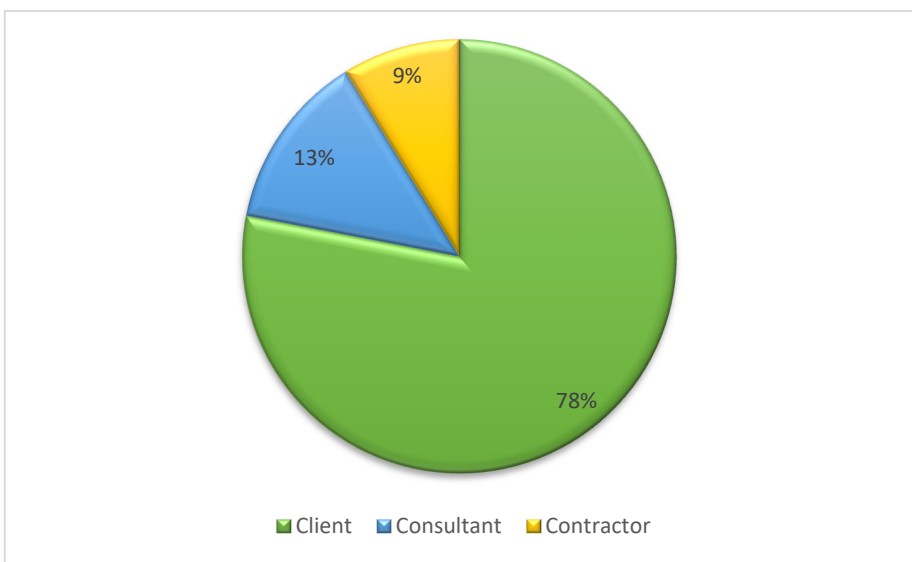

**Figure 10.** Major responsibility of stakeholders in decision-making process.

Figure 10 illustrates the major responsibility that falls on stakeholders in the latter stages of the decision-making process. According to the findings, the client plays an important part in the decision-making process. As a consequence, any delay caused by the customer is the most significant factor that contributes to delayed decision making in construction projects. It was noted that the process of decision making on the client's side is always extended. When clients take their time making decisions, it has a negative influence on the progression of the construction project, and it also has the ability to drive up both the cost and the length of the project. The client's ability to make sound decisions might be essential to the project's ability to accomplish the desired outcomes with high quality at the lowest possible cost and on schedule [43]. The client's inability to make timely decisions might result in a rise in costs, and the project might not be able to meet the requirements [73]. In a similar manner, the client has a vital part to play, despite the fact that the consultant and contractors both play essential roles in this crucial process.

In the following, Figure 11 shows the key factors of timely decision making in construction projects.

The four key factors observed in this study were (1) technical expertise, (2) incomplete documents, (3) lack of good leadership, and (4) lack of coordination/communication, because all these factors had RII scores of more than four.

The first important observed factor was technical expertise. In the construction industry, the lack of technical expertise in management to make timely decisions is a significant problem. Technical expertise is based on knowledge, information, data, assessment reports, frequent internal meetings, and regular visits to the construction site to get updates and inquire about the issues and factors that cause project delay [74,75]. The collective information can help to timely make decisions to resolve pending cases and maintain the flow of construction activities.

The second important factor is incomplete documents. It was observed that incomplete records create issues and hinder the progress of construction work. Incomplete sets of documents create problems during the construction of a project, which impacts timely decision making [69]. Effectively managed documents help stakeholders make timely decisions during construction by the specified time mentioned in the contract [76].

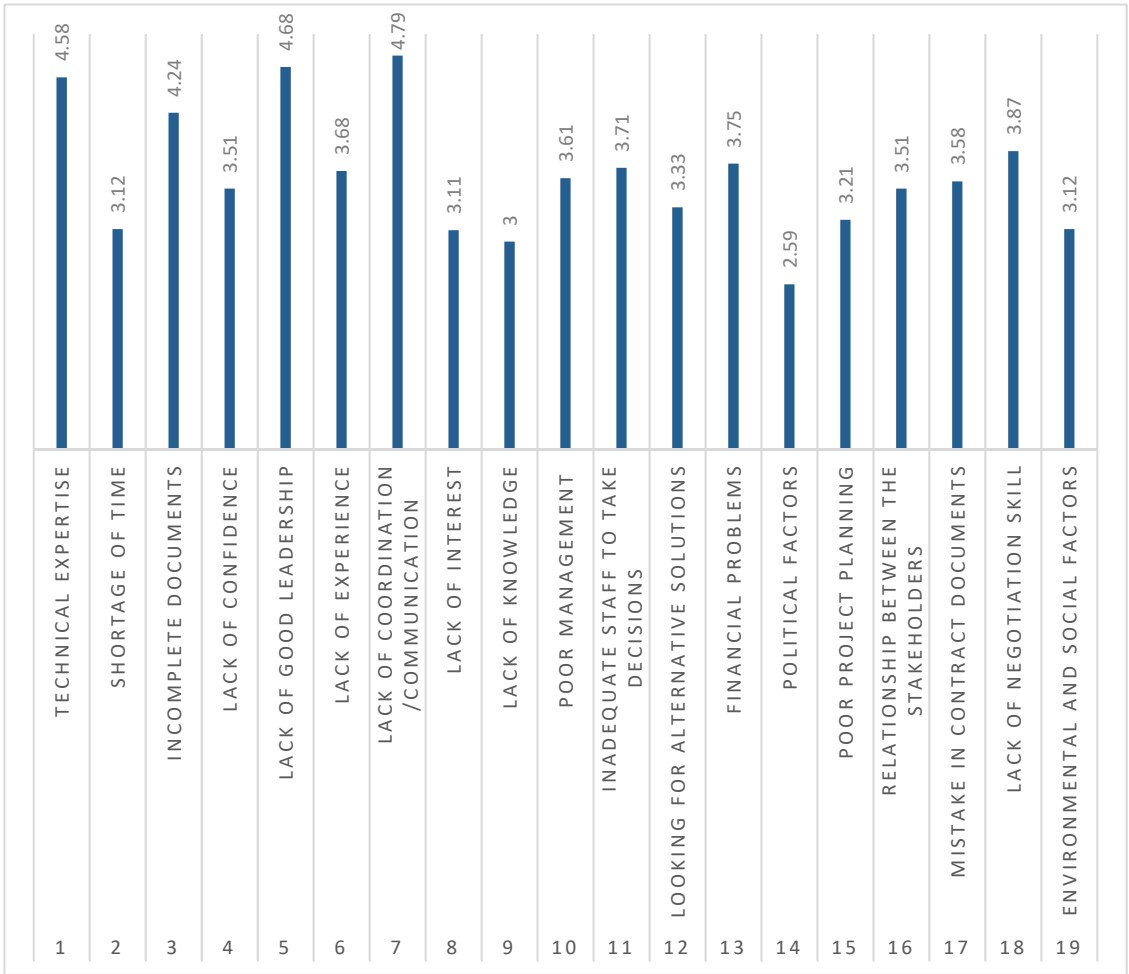

**Figure 11.** Factors affecting timely decision making.

The lack of good leadership stands as the third key factor. It was noticed that there is limited leadership who can make timely decisions to resolve and minimize issues in projects in the construction industry. The lack of management in the construction sector has been a significant issue faced by the construction sector. The construction industry should hire experienced staff/leadership that can take projects in the right direction [77]. Decision making in organizational leadership and management activities impacts creativity, growth, effectiveness, success, and goal accomplishment in organizations. There is a high need for change and improvement in organizational executives' decision making, including accommodating technology, diversity, globalization, policy, teamwork, and leadership effectiveness [78].

The fourth important factor affecting timely decision making is the lack of coordination. It was observed that there is a lack of coordination between stakeholders and project management. This lack of coordination impacts decision makers, and as a result, they fail to make timely decisions in construction projects. The lack of coordination affects project performance. Stakeholders should coordinate efforts throughout the project to make fair decisions to ensure the required goal timely achievement [79]. Introducing an effective communication tool could be helpful for decision makers on the project to communicate with each other [80].

The study results were validated using Cronbach's alpha to check the reliability of the collected data as shown in Table 3.

**Table 3.** Reliability analysis.

| Cronbach's Alpha | No. of Items |
|:---:|:---:|
| 0.88 | 91 |

The value of the reliability analysis of the 91 received questionnaires was 0.88, indicating good data fit. Thus, the respondent feedback was fit and normal.

In the following, suggestions and decision process improvement actions are investigated. The results of Phase 1 highlight that the client has a crucial role in timely decision making in construction projects. Proper and timely decision making is a fundamental action in any project. The client should be briefly and timely informed about the issues of the project so that the client can make timely decisions [81]. The client implements the decision-making process to be followed by all the stakeholders to make correct and timely decisions [82]. The government sector can implement the decision-making process to be followed by all stakeholders (contractors, consultants/engineers, and clients) in the construction sector. The process indicates the right path and directions to make the right decision at the right time, as shown in Figure 12.

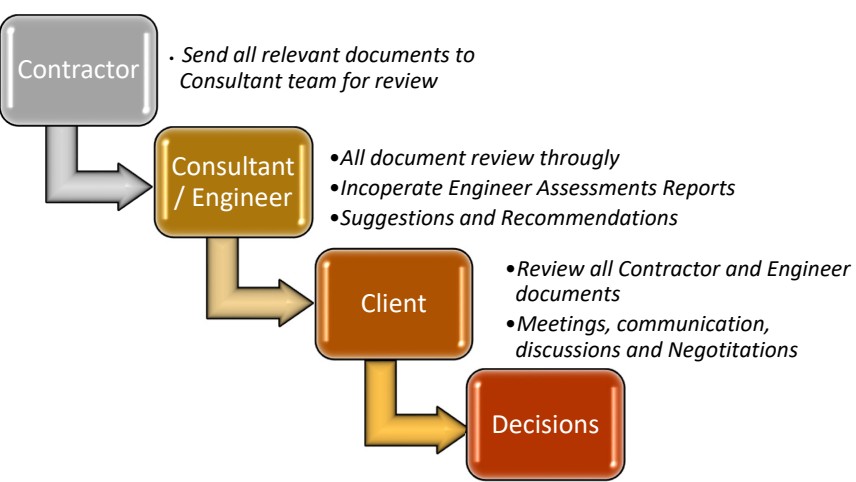

**Figure 12.** Process of decision making of client.

A model suggested for the client's decision making depends on the contractor's and consultant's set of evidence and documents provided to the client. The contractor and the consultant should provide the complete set of documents to the client for review. The client should coordinate with the stakeholders to discuss and negotiate the issues to quickly make decisions. Table 4 comprises the experts' suggestions for effective and timely decision making in construction projects analyzed in this study.

**Table 4.** Experts' suggestions.

| S. No. | Key Aspect | Suggestion |
|:---|:---|:---|
| 1 | Awareness | The client should be aware and be updated about every single issue happening at the construction site. |
| 2 | Interaction | The team should interact with all stakeholders working at the site and appropriately interact with the client and the contractor to help to make timely decisions. |
| 3 | Participation | The client team members should participate in each contractor and consultant meeting and seminar to be updated about valuable information. |
| 4 | Communication | Frequent communication avoids delays in the decision-making process. |
| 5 | Social skills | The client should manage and maintain relationships with working stakeholders with proper coordination. |

**Table 4.** *Cont.*

| S. No. | Key Aspect | Suggestion |
| --- | --- | --- |
| 6 | Tools | The proper use of tools and process helps in fair decision making. |
| 7 | Knowledge | Essential and massive knowledge help the client to make correct decisions. |
| 8 | Transparent decisions | The client should make bold and transparent decisions during construction. It helps and motivates the stakeholders to timely complete the project. |
| 9 | Decision model | The client should develop the joint decision model used by managers for setting a strategic direction, intelligently allocating resources, formulating tactical plans, and coordinating the resulting activity. |
| 10 | Decision-making process | The decision-making process should support stakeholders to make quick, right, and fair decisions. |

The recommendations made by the experts provide guidelines for an objective method for decision making. Making the appropriate choices at the appropriate times may assist all of the project stakeholders in accomplishing essential objectives. High construction progress and performance are guaranteed when there is proper communication and coordination between the employees who provide assistance and those who make decisions at the appropriate moment. Decision makers in a construction project should have extensive knowledge of the project and follow the decision-making process that has been suggested by the client or established by the key project stakeholders in an early stage in order to avoid delaying the progress of construction work. Based on the experts' feedback and results of this study, this research proposes a decision-making model as shown in Figure 13. The construction industry in Saudi Arabia in particular and other countries in general can follow this model to improve the decision-making process in mega construction projects.

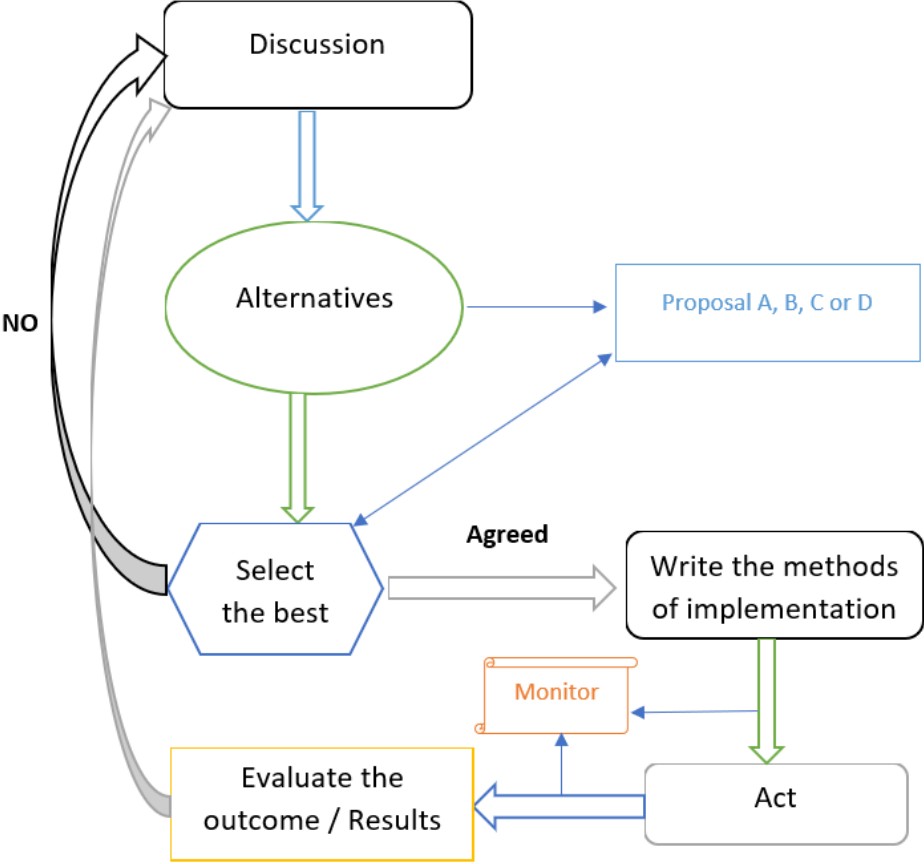

**Figure 13.** Proposed decision-making model.

Involved parties and stakeholders should adequately discuss problems before making any decisions. They should work together often, communicate openly, and understand one another in order to overcome the problems. The team should gather to evaluate the various solution benefits and drawbacks before settling on the best option for each issue. The implementation strategy should be written down and approved by all team members before it can be signed by all parties. The operation department should obtain the implementation strategy and execute it as necessary. Decision makers should regularly monitor the implementation process and evaluate the outcomes. Decision makers should discuss the findings and point out any small mistakes that need to be avoided or minimized in the decision-making process moving forward after successfully executing the plan and attaining the desired goals. The simple and straightforward procedure helps decision makers choose the best options using effective deliberation and communication. Collective stakeholder decision making and contribution could be advantageous for building projects. It could aid in lowering the likelihood of anticipated delays and help projects avoid numerous issues.

## 7. Validation of the Proposed Model

Validating such models is usually a challenge; however, this model was verified with focused interviews with top industry professionals working on big infrastructure projects. The professionals offered their comments on the usefulness of the model, and those comments were included into the final version of the decision-making model, which is shown in Figure 13. The specialists stressed the need to use a proposal-based strategy for projects, as well as early meetings and integration during projects, as a way to reach better judgments in a timely manner, particularly with regard to big infrastructure projects in Saudi Arabia. The dynamics of the nation are somewhat distinct from those of the other countries in the area, and the suggested model could be fairly effective in the event that it is used in projects.

## 8. Conclusions

Construction activity is impacted by stakeholders' late decision making in mega projects due to project complexity. This research determined that the client plays a crucial role in decision making and its timely implementation. The primary characteristics that influence the immediacy of decision making discovered in this research include (1) technical expertise, (2) inadequate documentation, (3) lack of competent leadership, and (4) lack of coordination, among others. The conclusion is that the customer should take the required measures and procedures to prevent decision-making delays. The attitude used toward the customer should be one that is proactive, collaborative, and active. It was proposed that the main stakeholders of the project create a decision-making process in the early stage of the project in order to enhance the performance of decision making. Failure to do so may result in a variety of issues in the project, such as an increase in overall risk, longer completion time, more costs, and worse quality. The correct use of the decision-making process in the construction industry enables the many parties involved to reach just and timely conclusions that are in the best interest of the building project. In order to make the most informed decisions possible, the client is responsible for following and repeating the process as well as staying updated on relevant topics. The client's ability to make timely decisions results in time and money savings in projects. That is an encouraging sign and demonstrates the client's interest in seeing initiatives through to a successful conclusion.

## 9. Recommendations

Based on the results, this research recommends the following for better and timely decision making in construction projects:

- Every construction project should design a decision-making process with specific tasks for each stakeholder.

- The client should maintain a strong relationship and coordinate with the consultant and the contractor to monitor project problems and progress. The client should avoid late decision making to improve project performance.
- The complete and proper set of project documents should be prepared and made available to key stakeholders. Stakeholders should frequently meet, share problems, and discuss possible solutions.
- Technical experts with vast field knowledge and experience should be hired to accomplish complex actions in projects, tasks, and processes.
- Training programs should be initiated to improve thinking capabilities, proper communication and coordination, strong leadership, and problem-solving techniques.

## 10. Limitation of the Study

This study focuses on mega projects in Saudi Arabia, such as rail projects, BRT projects, and airport projects. The research work is limited to companies involved in mega infrastructure projects in the Saudi Arabian construction industry.

## 11. Significance of Study

This research could support experts working in mega infrastructure projects in the Saudi Arabian construction industry to improve decision-making practices in projects. This research proposes a decision-making model that should be adopted to enhance the decision-making process. The key factors highlighted in this research should be considered to avoid delays in decision making in projects.

## 12. Novelty of the Research

This research is different from existing work. No similar attempts have been made focusing on the Saudi Arabian construction industry. The country has high potential in this industry due to the announcement of mega infrastructure projects to meet Saudi Vision 2030. Special attention was given to factors of delays in the decision-making process. The country' governance protocols are different from those of other countries; thus, the existing decision-making delay factors and processes are not very effective.

**Author Contributions:** Methodology, B.S. and Q.H.K.; Formal analysis, Q.H.K.; Investigation, N.Y.Z.; Resources, B.S.; Writing—original draft, H.H.S. and B.S.; Writing—review & editing, S.H.K.; Project administration, N.Y.Z. All authors have read and agreed to the published version of the manuscript.

**Funding:** This research received no external funding.

**Institutional Review Board Statement:** Not applicable.

**Informed Consent Statement:** Not applicable.

**Data Availability Statement:** Not applicable.

**Acknowledgments:** The authors are thankful to Prince Sultan University Riyadh for expert support and for paying the article processing charges (APC) for this paper.

**Conflicts of Interest:** The authors declare that they have no conflict of interest regarding the research.

## Appendix A

**SURVEY QUESTIONNAIRE**
**Delay in Decision-Making Affecting Construction Projects.**
The purpose of this questionnaire is to determine the factors affecting the delay in decision-making process in the mega infrastructure of Saudi Arabian construction industry.

All information and detail provided in the questionnaire are kept **CONFIDENTIAL** and served as an important guide for this research purpose only.

Your precious time and attention to provide the valuable answer for this questionnaire are appreciated. Thank you very much.

PHASE: 1

**SECTION I: Demographics (please fill in the blanks or appropriate check box)**

(1)   Name of Organization: ____________________________________________

(2)   Organization's Address: ____________________________________________

(3)   Your current Position: ____________________________________________

(4)   Please specify type of your current organization:

| | |
|---|---|
| Consultant | Contractor (Category): _____________________ |
| Client | Others, please specify, _____________________ |

(5)   Please specify your total working experience

| | |
|---|---|
| Less than 5 years | 6–10 Years |
| 11–15 Years | Above 15 Years |

(6)   Please specify your total working experience in fast track mega infrastructure projects

| | |
|---|---|
| Less than 5 years | 6–10 Years |
| 11–15 Years | Above 15 Years |

Name (Optional): _____________________________________ Date: _____

E-mail address (Optional): _______________________

Contact Number (Optional): _______________________

<u>**SECTION II: (please circle or tick the appropriate scale value in the given boxes)**</u>

| which stakeholder have major responsibility to take timely decisions in mega infrastructure Projects of Saudi Arabian Construction Industry? | 1<br>Client | 2<br>Consultant | 3<br>Contractor | 4<br>Sub-contractor |
|---|---|---|---|---|

<u>**SECTION III: (please circle or tick the appropriate scale value in the given boxes)**</u>
Please choose/select the importance factor that affecting the Delay in Decisions-Making

| Factor Affecting the Delay in Decision-Making Process in the Construction Industry | Scale | | | | |
|---|---|---|---|---|---|
| | Very Important | Important | So-So | Less Important | Not Important |
| 1   Technical Expertise | 1 | 2 | 3 | 4 | 5 |
| 2   Shortage of time | 1 | 2 | 3 | 4 | 5 |
| 3   Incomplete documents/evidence | 1 | 2 | 3 | 4 | 5 |
| 4   Lack of confidence | 1 | 2 | 3 | 4 | 5 |
| 5   Lack of good leadership | 1 | 2 | 3 | 4 | 5 |
| 6   Lack of experience | 1 | 2 | 3 | 4 | 5 |
| 7   Lack of coordination/communication | 1 | 2 | 3 | 4 | 5 |
| 8   Lack of Interest | 1 | 2 | 3 | 4 | 5 |
| 9   Lack of knowledge | 1 | 2 | 3 | 4 | 5 |
| 10   Poor management | 1 | 2 | 3 | 4 | 5 |
| 11   Inadequate staff to take decisions | 1 | 2 | 3 | 4 | 5 |
| 12   Looking for alternative Solutions | 1 | 2 | 3 | 4 | 5 |
| 13   Financial Problems | 1 | 2 | 3 | 4 | 5 |
| 14   Political factors | 1 | 2 | 3 | 4 | 5 |
| 15   Poor project planning | 1 | 2 | 3 | 4 | 5 |

| Factor Affecting the Delay in Decision-Making Process in the Construction Industry | | Scale | | | | |
|---|---|---|---|---|---|---|
| | | Very Important | Important | So-So | Less Important | Not Important |
| 16 | Relationship between the stakeholders | 1 | 2 | 3 | 4 | 5 |
| 17 | Mistake in contract documents | 1 | 2 | 3 | 4 | 5 |
| 18 | Lack of negotiation skill | 1 | 2 | 3 | 4 | 5 |
| 19 | Environmental and social factors | 1 | 2 | 3 | 4 | 5 |

**PHASE II:** The author is requesting experts to recommend suggestion based on the factor mention that could be helpful during Decision-Making Process.

| | Factors | Expert Suggestion |
|---|---|---|
| 1 | Awareness | |
| 2 | Interaction | |
| 3 | Participation | |
| 4 | Communication | |
| 5 | Social skills | |
| 6 | Tools | |
| 7 | Knowledge | |
| 8 | Transparent decisions | |
| 9 | Decision Model | |
| 10 | Decision-Making process | |

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
