# Peer review of "Delay in Decision-Making Affecting Construction Projects: A Sustainable Decision-Making Model for Mega Projects"

_sustainability, doi:10.3390/su15075872_

Round 1

Reviewer 1 Report

Observations:

1.  Text line: 86: There is no such previous study conducted... - Maybe it's better: To the best of the author's knowledge... (Like this statement is too narrow, indicating an overly confident attitude towards the paper. I highly doubt that they've actually analyzed all the available papers to acknowledge this.)

2.  Text line: 89-95: Too much repetition, this should be formulated in 1 sentence, avoiding using too many "suggests"

3.  Text line: 97: Correction to The decision-making process (in line 99 the nomenclature doesn't initially align)... based on various factors (I would add: such as and name a few to exemplify  which are the factors)

4.  Text line: 99-102: Not understandable / illogical statement. (Various people - Who are they? Should be written - stakeholders)

5.  Text lines: 102-103: How come all of a sudden speaking of issues? The statement doesn't fit here. 

6.  Text lines: 104-105:  lack of qualified persons. (Should have said: Lack of skilled employees / workforce...) + Statement based on what?

7.  Text line: 106: What is collective information? Wrong word usage.

8.  Text line: 108: Misses Figure 1. which would lead the reader to the image beyond. + What is the source of the image?

9.  Text line: 112: Repetition of line 108.

10.  Text line: 114-127: Unclear repetitive statements, using poor vocabulary to depict the same notion. 

11.  Text lines: 126 - 127: Says there are multiple researchers, but doesn't give any example, but later shows scarcely only one example.

12.  Text lines: 146 - 166: Introduces various terms without prior introduction. Additionally lacks literature sources; uses poor vocabulary and lack standardization especially in Figure 3 (misses the full stop).

13.  Text lines: 169 - 177: In the previous section talks about reconstruction projects and in this one about construction projects.

14.  Text lines: 200 - 208: Again talks about the decision-making process. 

15.  Text lines: 209 - 221: Again talks about the renovation. At this point consistency is lost causing a lot of confusion regarding whether it is about construction or reconstruction projects, additionally talking about energy efficiency. 

16.  Text lines: 279 - 281: Again talks about the decision-making process, and vast literature regarding the subject (which hasn't been presented in such a glamorous way as the author states it has)

17.  Text line: 290 - 310: Research Methodology - Completely wrongfully written. Lack of data sources, why he used the selected method, constantly relying on personal experience.

18.  Text lines: 334 - 372: The author should consider bundling certain results and clustering them into groups due to a high variety of observed factors. Additionally, the author should work on a better graphical representation of the results. And should provide us with a global intersection of the data subject's characteristics. 

19.  Text line: 378 - 319: How did the author arrange these factors? Based on what? As I understood, the selection criteria was a result above 4, but even in that case, the author failed to address all of the issues missing out "Incomplete documentation".

20.  Text line: 420 - 425: Tabular representation is too scarce. The author should include more data to provide us with scales and give a reader a sense of results acceptability. Like this, it is not understandable.  

21.  Text line: 426 - 438: Is this the beginning of the new research topic? 

22.  Text line: 439 - Figure 12 is completely misplaced. Shouldn't be placed here, but in the section regarding the literature review.  

23.  Text line: 460 - 464 - Misses literature sources. 

24.  Text line: 487 - 497 - The author should rewrite this section entirely. 

25.  Text line: 499 - 520 - The author should rewrite this section entirely.

Overall impression: 

The papers lack consistency. The vocabulary used is highly repetitive and doesn't meet the proposed research demands. Additionally, the author misses adequate literature sources to confirm/deny the statements throughout the paper. The author failed to clearly address the purpose, problem, and the scope of research resulting in poor results representation and perhaps statistical modeling. The author should work on vocabulary consistency to use standardized nomenclature and avoid potential mistakes or misinterpretations. 

Overall, poorly written article, primarily lacking the right ways of expression, literature, consistency, standardization, and the right way to interpret and present the research results.

Author Response

Dear Sir

First of all, let me thank you for your valuable and constructive suggestions to improve the paper. We did our best to address each comment in detail. Almost, we revised the whole paper and the language is also improved. We hope the revised version will meet your expactations. 

Thanks again

Author Response

Dear Sir
First of all, let me thank you for your valuable and constructive suggestions to improve the paper. We did our best to address each comment in detail. Almost, we revised the whole paper and the language is also improved. We hope the revised version will meet your expectations. 
Thanks again

Reviewer 3 Report

This paper presents a sustainable decision making model for mega projects

1- The introduction section is short and it seems that it should be more complete. The structure of the article should be specified in the introduction section.

2- Literature review section should be added in the article and the articles of recent years should be mentioned in them.

3- Research methodology should be presented schematically.

4- The method of validating the intended methodology is very short and must be completed. How is the model approved now?

5- The conclusion is very short and unclear. Innovation is not transparent paper.

6- If the results of the work are compared with previous studies, a better result can be obtained.

Author Response

(The authors gave the same response as above.)

Round 2

Reviewer 2 Report

Thank you!